# Penetration Fracture Mechanism of Tungsten-Fiber-Reinforced Zr-Based Bulk Metallic Glasses Matrix Composite under High-Velocity Impact

**DOI:** 10.3390/ma16010040

**Published:** 2022-12-21

**Authors:** Chengxin Du, Feng Zhou, Guangfa Gao, Zhonghua Du, Huameng Fu, Zhengwang Zhu, Chun Cheng

**Affiliations:** 1School of Mechanical Engineering, Nanjing University of Science & Technology, Nanjing 210094, China; 2Institute of Metal Research, Chinese Academy of Sciences, Shenyang 110016, China; 3Impact and Safety Engineering, Ningbo University, Ningbo 315211, China

**Keywords:** penetration fracture mode, tungsten-fiber-reinforced Zr-based bulk metallic glass matrix composite (WF/Zr-MG), impact velocity, bending and backflow

## Abstract

In order to adapt to the launch velocity of modern artillery, it is necessary to study the fracture mechanism of the high-velocity penetration of penetrators. Therefore, the penetration fracture mode of tungsten-fiber-reinforced Zr-based bulk metallic glass matrix composite (WF/Zr-MG) rods at a high velocity is studied. An experiment on WF/Zr-MG rods penetrating into rolled homogeneous armor steel (RHA) was carried out at 1470~1650 m/s. The experimental results show that the higher penetration ability of WF/Zr-MG rods not only results from their “self-sharpening” feature, but also due to the fact they have a longer quasi-steady penetration phase than tungsten alloy (WHA) rods. Above 1500 m/s, the penetration fracture mode of the WF/Zr-MG rod is the bending and backflow of tungsten fibers. Our theoretical calculation shows that the deformation mode of the Zr-based bulk metallic glass matrix (Zr-MG) is an important factor affecting the penetration fracture mode of the WF/Zr-MG rod. When the impact velocity increases from 1000 m/s to 1500 m/s, the deformation mode of Zr-MG changes from shear localization to non-Newtonian flow, leading to a change in the penetration fracture mode of the WF/Zr-MG rod from shear fracture to the bending and backflow of tungsten fibers.

## 1. Introduction

The tungsten-fiber-reinforced bulk metallic glass matrix composite (WF/Zr-MG) is an ideal material for penetrators in armor-piercing projectiles because of its high strength and high density. The compressive yield strength of WF/Zr-MG is higher than 2.0 GPa, and the density of WF/Zr-MG is around 17.1 g/cm^3^ when the volume faction of WF is 80% [1,2,3,4,5,6,7,8]. In addition to the high strength and high density, some research results show that WF/Zr-MG rods exhibit a “self-sharpening” feature during the penetration process, similar to depleted uranium alloy [9,10,11,12,13,14]. The penetration ability of WF/Zr-MG rods is more than 10% higher than that of WHA rods under similar conditions [1,2,3,10,11,12,13,14,15,16]. Compared with the other “self-sharpening” materials, the WF/Zr-MG rod also exhibits better penetration ability [17,18]. Because the “self-sharpening” feature results from the penetration fracture mode, the fracture mode of the penetration process should be studied to explore the mechanism behind the “self-sharpening” of WF/Zr-MG rods.

Guo et al. [14] studied the fracture mode of WF/Zr-MG rods penetrating a carbon steel target at a velocity of 653 m/s. Rong et al. [16] studied the fracture mode of WF/Zr-MG rods penetrating 30CrMnMo at a velocity of 804~1076 m/s. Under these conditions, the penetration fracture mode of WF/Zr-MG rods is similar to that of depleted uranium alloy. The head of the WF/Zr-MG rod is conical, with no obvious expansion. Chen et al. [12] studied the penetration fracture mode of the WF/Zr-MG rod at 1300 m/s. The research results of Chen et al. proved that the penetration fracture mode of WF/Zr-MG rods is shear fracture at high velocity, but at this point the head of the rod expands. Magness et al. [19] also found that the head of the WF/Zr-MG rod expanded when penetrating RHA. The results show that the penetration fracture mode of the WF/Zr-MG rod is related to the impact velocity. However, with the development of military technology, projectiles are being launched faster. The velocity of most armor-piercing projectiles is more than 1500 m/s. However, the current research on the penetration fracture mode of WF/Zr-MG rods is below 1500 m/s. It cannot reveal the penetration mechanism of WF/Zr-MG rods at high velocity. Therefore, we carried out an RHA penetration experiment to study the penetration fracture mode of WF/Zr-MG rods at a velocity above 1500 m/s. The influence of the impact velocity on the penetration fracture mode of WF/Zr-MG rods is revealed.

## 2. Experiment on WF/Zr-MG Rod

### 2.1. Setup of Experiment

The matrix metal glass phase of WF/Zr-MG used in the experiment is Zr_41.25_Ti_13.75_Ni_10_Cu_12.5_Be_22.5_ (called Zr-MG in this paper). Moreover, Ø0.3 mm tungsten fibers are used and homogenously embedded in the matrix. Zr-MG is prepared by the constitutive elements with purity of 99.5 pct or higher. The elements are combined and placed in an induction furnace, and then heated to prepare the Zr-MG ingots. The tungsten fibers are straightened to ensure parallel arrangement. Acetone and ethanol are used to clean the tungsten fibers, to enhance the bonding strength between the tungsten fibers and the matrix. Tungsten fibers are unidirectionally arranged in the sealed end of an evacuated quartz tube. Then, the Zr-MG ingots are melted and infiltrated into the into quartz tube by high pressure. The quartz tube containing tungsten fibers and Zr-MG is rapidly solidified to prepare WF/Zr-MG. Figure 1 shows a portion of the cross-section the WF/Zr-MG.

The projectile and the rod used in the experiment are shown in Figure 2. Figure 2a is an expanded view of a projectile, including a block, a platen, three sabots and a penetrator. A steel platen is placed between the penetrator and the block, and the platen and the block are used to push the penetrator. The sabots are used to fit the larger-diameter smooth bore to fire a smaller penetrator. Figure 2b shows the penetrator. The penetrator includes a tail, a nose and a rod. The tail and the nose are installed in the bottom and top of the rod. The diameter and the length of the rod are Ø10 mm and 54 mm, respectively.

Projectiles are launched by Ø25 mm smooth bore artillery. The setup of experiments is shown in Figure 3. An RHA target with a thickness of 80 mm is placed 10 m in front of the artillery. A laser doppler anemometer is used to measure the velocities of the rods. The impact velocities of the rods vary from 1470 m/s to 1650 m/s.

A total of 4 projectiles were used. Two of the rods were made of WF/Zr-MG. The other two rods were made of WHA to compare the penetration behavior of WF/Zr-MG and WHA. The masses of the projectiles and rods are shown in Table 1.

### 2.2. Experimental Results and Analysis

Figure 4 and Figure 5 show the craters penetrated by the WF/Zr-MG rods and WHA rods, respectively. As shown in both figures, all the craters are straight, without ballistic deflection. The crater can be divided into three parts. The crater diameter of the first part increases gradually, the crater diameter of the second part remains unchanged, and the crater diameter of the third part decreases gradually. The three parts of the crater correspond to the transient entrance phase (TEP), the quasi-steady penetration phase (QSPP), and the secondary penetration phase (SPP) of the penetration process, respectively.

Table 2 shows the dimensions of the crater, including the penetration depth and the lengths of three parts of the crater. As shown in Table 2, the penetration depths of WF/Zr-MG rods are higher than those of WHA rods, which proves that the penetration ability of WF/Zr-MG is higher than that of WHA. The length of the first part and third part of the crater penetrated by WF/Zr-MG rods is lower than that of the crater penetrated by WHA rods, which proves that the transient entrance phase and the secondary penetration phase of the penetration process of WF/Zr-MG rods are shorter than those of WHA rods. The results show that the WF/Zr-MG rods have a longer retention time in the quasi-steady penetration phase, so the WF/Zr-MG rods will perform better when the rod penetrates with a larger aspect ratio. Furthermore, Table 2 shows the average diameter of the craters in the quasi-steady penetration phase. The average diameter of the quasi-steady penetration phase of the crater penetrated by WF/Zr-MG rods is smaller than that of the crater penetrated by WHA rods, highlighting the “self-sharpening” feature of WF/Zr-MG rods. The experimental results show that the higher penetration ability of WF/Zr-MG rods not only results from the “self-sharpening” feature, but also from a longer quasi-steady penetration phase than the WHA rod.

## 3. Penetration Fracture Mode of WF/Zr-MG Rod

It can be seen from Table 2 that the secondary penetration phase of the WF/Zr-MG rod is very short, so it is in the quasi-steady penetration phase until the end of penetration. Therefore, the fracture morphology of the residue rod at the bottom of the crater can represent the fracture morphology of the rod during penetration.

Figure 6 is a metallographic photo of the residual WF/Zr-MG rods (1523 m/s) obtained in this experiment; Figure 7 is a metallographic photo of the residual WF/Zr-MG rods (811.2 m/s) obtained from Rong et al. [16]. Figure 6 shows that the crater bottom penetrated by WF/Zr-MG rods at 1500 m/s is spherical, and the tungsten fibers at the crater bottom flow to both sides of the crater. In Figure 7, there is an obvious shear band on the head of the residual WF/Zr-MG rods at 811 m/s, which proves that the penetration fracture mode of WF/Zr-MG rods in Figure 7 is shear fracture. By comparison, it can be found that the penetration fracture mode of the WF/Zr-MG rods in Figure 6 is the bending and backflow of tungsten fibers.

As seen in Figure 4, many residual penetrator materials remain in the crater after the penetration of the WF/Zr-MG rod. The residual materials in the crater include aluminum alloy (from the tail) and tungsten fibers. There is little Zr-MG attached to the residual tungsten fibers. The tungsten fibers in Figure 4 are longer than the fractured tungsten fibers in Figure 7, similar to the experimental results of the bending and backflow of tungsten fibers obtained before, as seen in Figure 8 [20].

In order to further analyze the penetration fracture mode of WF/Zr-MG rods at a high impact velocity, electron microscopy was used to analyze the interface of the rod and target. Figure 9a is an SEM image of the interface inside the red dashed box in Figure 6, and Figure 9b shows the composition of the material in Figure 9a. As seen in Figure 9b, Zr and Mn are the main materials attached to the target on the interface. The Zr and Mn attached to the target come from the matrix in the rod, which proves that the matrix in the rod melts during the penetration process. The melting of the matrix results in the loss of the constraint on the tungsten fibers, so that the tungsten fibers bend in a free state and backflow to the rear of the crater with the penetration, remaining in the crater. Therefore, it is proven that the penetration fracture mode of the WF/Zr-MG rod is the bending and backflow of tungsten fibers when the impact velocity is higher than 1500 m/s.

## 4. Discussion

### 4.1. Influence Factors of the Fracture Mode of WF/Zr-MG

The above test results have proven that the penetration fracture mode of WF/Zr-MG rods changes from shear fracture to the bending and backflow of tungsten fibers at velocities above 1500 m/s. In order to further analyze the transition conditions of the fracture mode, the deformation mode of WF/Zr-MG was analyzed.

In our previous study, the dynamic compression deformation mode of WF/Zr-MG at room temperature was shear fracture, as shown in Figure 10a [5]. However, the study of Chen et al. shows that the dynamic compression deformation mode of WF/Zr-MG changes to the bending of tungsten fibers at high temperatures, as shown in Figure 10b [4]. As the test temperature used in the work of Chen et al. was 773 K, much lower than the melting point of tungsten fibers, the deformation mode of WF/Zr-MG transformed due to the softening of the matrix of Zr-MG. Therefore, the main factors affecting the fracture mode of WF/Zr-MG are the strength and deformation mode of the matrix of Zr-MG.

Previous dynamic compression tests of WF/Zr-MG have shown that the stress propagation is also a factor affecting its deformation mode [5]. However, in the process of penetration, the impact velocities and impact energy are much higher than those of the dynamic compression test, which directly leads to the shear failure of WF/Zr-MG. Therefore, the propagation of stress waves is no longer an important influence factor in the penetration fracture mode.

### 4.2. Influence Mechanism of the Matrix of Zr-MG on Penetration Fracture Mode of WF/Zr-MG

Lu et al. [21] studied the compression deformation mode of Zr-MG, and the results showed that, under different temperatures and strain rates, the matrix of Zr-MG presented three distinct deformation modes: shear localization, non-Newtonian flow, and Newtonian flow, as shown in Figure 11. Lu et al. also obtained the boundaries between the three distinct deformation modes of Zr-MG from their investigation, as shown in Figure 12 [21].

According to the results, the boundary from shear localization deformation to non-Newtonian flow deformation was obtained by fitting.
(1)$$\frac{\dot{\varepsilon }}{\dot{\bar{\varepsilon }}}=1.3\times 10^{-4}\times \exp\left[51.15{\left(\frac{T}{T_{room}}\right)}^{2}-185\frac{T}{T_{room}}+167\right]$$
where $\dot{\bar{\varepsilon }}=1\;s^{-1}$ and $T_{room}=298\;K$.

The boundary from non-Newtonian flow and Newtonian flow can be calculated using the viscosity ratio η and Newtonian viscosity $\eta_{n}$ [21],
(2)$$\frac{\eta }{\eta_{n}}=1-\exp\left[-{\left(\frac{\alpha }{\dot{\varepsilon }\eta_{n}}\right)}^{\beta }\right]$$
where $\dot{\varepsilon }$ is strain rate, α and β are fitting parameters for the experimental data. For Zr-MG, the values of α and β are 172 MPa and 0.85 respectively [21,22].

According to the modification of the free volume model by Cohen and Grest, the relationship between Newtonian viscosity and the temperature of materials can be obtained [22,23]:(3)$$\eta_{n}=\eta_{0}\exp\left[\frac{2bv_{m}\varsigma_{0}/k_{B}}{T-T_{0}+\sqrt{{\left(T-T_{0}\right)}^{2}+\frac{4v_{a}\varsigma_{0}}{k_{B}}T}}\right]$$
where $\eta_{0}$ is a pre-factor, $bv_{m}$ is a critical volume for flow, $\varsigma_{0}$, $k_{B}$ and $v_{a}$ are material constants. For Zr-MG used in this experiment, $\eta_{0}=4\times 10^{-5}\mathrm{Pa}\cdot s$, $bv_{m}\varsigma_{0}/k_{B}=4933K$, $T_{0}=672K$, $v_{a}\varsigma_{0}/k_{B}=40.5K$ [22].

Since the matrix of Zr-MG is the main factor affecting the deformation mode of WF/Zr-MG, the fracture mode of WF/Zr-MG can be calculated indirectly using the deformation mode of Zr-MG. Therefore, the fracture mechanism of WF/Zr-MG rods during penetration can be calculated using the temperature and strain rate of Zr-MG.

High pressure of more than 10 GPa will be generated during the penetration process, which is far higher than the yield strength of the target and rods. Therefore, the interface of the target and rods can be treated as a fluid state during penetration. Then, the problems in the penetration process can be solved by the 1D shock wave theory of Rankine and Hugoniot and equations of state. The Rankine–Hugoniot equations are as follows [24]: (4)$$\left\{\begin{array}{l} \rho (U_{s}-U_{p})=\rho_{0}U_{s}\\ \rho_{0}U_{s}U_{p}=P-P_{0}\\ e-e_{0}=\frac{1}{2}(P+P_{0})(V_{0}-V) \end{array}\right.$$

The equation of state of solids is fit as follows: (5)$$U_{s}=C_{0}+SU_{p}$$
where ρ is the density, $\rho_{0}$ is the density under zero pressure; $U_{s}$ is the velocity of compressive shock wave; $U_{p}$ is the particle velocity in the compressed region; P is the pressure, $P_{0}=0$; e is the specific internal energy of the material, $e_{0}$ is the specific internal energy under zero pressure; V=1/ρ, it is the specific volume, $V_{0}$ is the specific volume under zero pressure; $C_{0}$ is the sound velocity; S is material constant.

According to the law of continuity [25],
(6)$$P_{t}=P_{p}$$

Before impacting the target, the velocity of all particles in the rod is v. During impact, the particle velocity in the compression zone will decrease $U_{p}$, and the velocity of all particles in the in the compressed region of the rod is $v-U_{p,p}$. Similarly, the particles in the compressed region of the target acquire a velocity of $U_{p,t}$ [25]. According to the law of continuity,
(7)$$U_{p,t}+U_{p,p}=v$$
where $U_{p,t}$ is the particle velocity in the compression zone of the target, and $U_{p,p}$ is the particle velocity in the compression zone of the rod.

According to Equations (4)–(7), the impact pressure on the projectile target interface under different impact velocities can be obtained:(8)$$\left\{\begin{array}{l} P_{p}=P_{t}=\rho_{t}(C_{0t}+S_{t}U_{pt})U_{pt}\\ U_{pt}=\frac{-b\pm \sqrt{b^{2}-4ac}}{2a} \end{array}\right.$$
where $a=\rho_{t}S_{t}-\rho_{p}S_{p}$, $b=\rho_{t}C_{0t}+2\rho_{p}S_{p}v+\rho_{p}C_{0p}$, $c=-\rho_{p}\left(C_{0p}v+S_{p}v^{2}\right)$.

During high-velocity impact, the high pressure of impact will cause a temperature rise in the materials. The relationship between the temperature rise and pressure can be calculated by the first and second laws of thermodynamics and Equations (4) and (5) [25]:(9)$$\frac{dT}{dP}=\frac{1}{2C_{v}}\left[\left(V_{0}-V\right)+\left(P-P_{0}\right)\frac{dV}{dP}-2T\frac{\gamma C_{v}}{V}\frac{dV}{dP}\right]$$
where T is the temperature; $C_{v}$ is the specific heat; $\gamma =\gamma_{0}\frac{V}{V_{0}}$, is the Grüneisen constant, $\gamma_{0}$ is the Grüneisen constant under zero pressure.

Table 3 shows the shock and thermodynamic properties of RHA, WF/Zr-MG, Zr-MG, and tungsten fiber. Using the parameters in Table 3, and with Equations (8) and (9), the relationship between the temperature rise and impact velocity during penetration can be calculated, as shown in Figure 13.

Anderson et al. [27,28] studied the strain rate of long rods penetrating a target at high velocity. This research shows that the strain rate of a WHA rod penetrating RHA is around 10^5^ s^−1^ at the artillery velocity. Therefore, at the impact velocity of 1000 m/s, the strain rate and temperature of Zr-MG in the WF/Zr-MG rod are 10^5^ s^−1^ and 435 K. Under this condition, the deformation mode of Zr-MG corresponds to shear localization. Therefore, the penetration fracture mode of the WF/Zr-MG rod is shear fracture at 1000 m/s. When the impact velocity increases to 1500 m/s, the temperature of Zr-MG in the WF/Zr-MG rod is 697 K, and its fracture mode will be non-Newtonian flow. The Zr-MG with non-Newtonian flow deformation loses the constraint on the tungsten fibers, so the penetration fracture mode of WF/Zr-MG rods changes to the bending and backflow of the tungsten fibers.

## 5. Conclusions

In this paper, the penetration fracture mode of WF/Zr-MG rods at 1500 m/s is analyzed by a penetration experiment. The craters and residual rods are used to analyze the penetration fracture mode. The penetration fracture mechanism of WF/Zr-MG rods is revealed using theoretical calculations. The conclusions obtained in this paper are as follows:
(1)When the impact velocity increases from 1000 m/s to 1500 m/s, the penetration fracture mode of WF/Zr-MG rods changes from shear fracture to the bending and backflow of tungsten fibers. The penetration fracture mode of WF/Zr-MG rods is mainly affected by the deformation mode of Zr-MG. At 1000 m/s, the fracture mode of Zr-MG is shear localization, and Zr-MG and tungsten fibers deform together, which leads to the penetration fracture mode of WF/Zr-MG rods being shear fracture. At 1500 m/s, the deformation mode of Zr-MG changes to non-Newtonian flow, and Zr-MG loses the constraint on the tungsten fibers, which causes the penetration fracture to change to the bending and backflow of tungsten fibers.(2)Under the same penetration conditions, the length of the quasi-steady penetration phase of WF/Zr-MG rods is longer than that of WHA rods, and the average crater diameter in the quasi-steady penetration phase is smaller, which results in the higher penetration ability of WF/Zr-MG rods.

## Figures and Tables

**Figure 1 materials-16-00040-f001:**
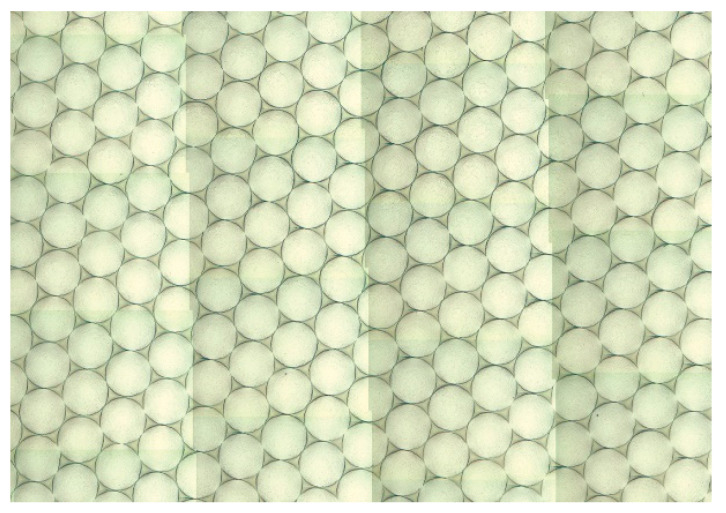
Portion of cross-section of WF/Zr-MG.

**Figure 2 materials-16-00040-f002:**
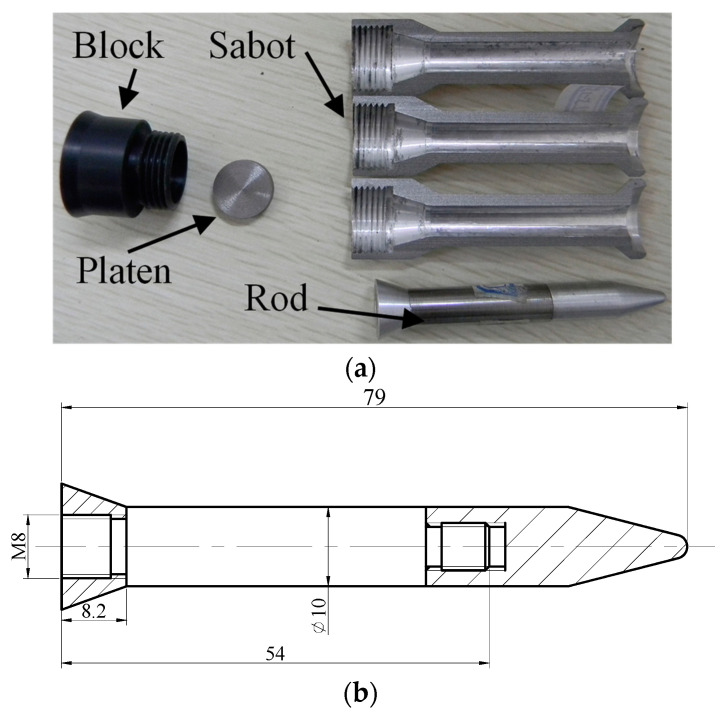
(**a**) Expanded view of projectile and (**b**) dimensions of rod.

**Figure 3 materials-16-00040-f003:**
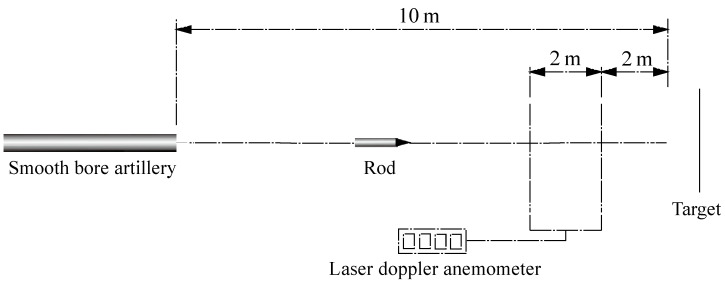
Setup of experiments.

**Figure 4 materials-16-00040-f004:**
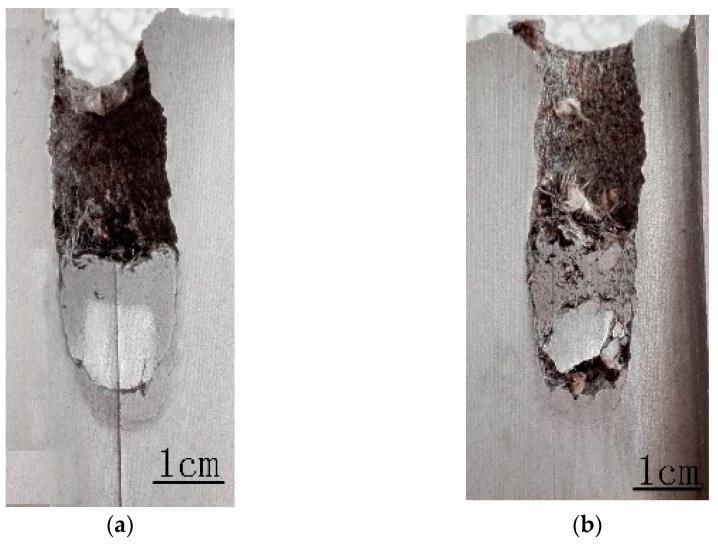
Longitudinal sections of craters penetrated by WF/Zr-MG rods. (**a**) *v* = 1560 m/s; (**b**) *v* = 1523 m/s.

**Figure 5 materials-16-00040-f005:**
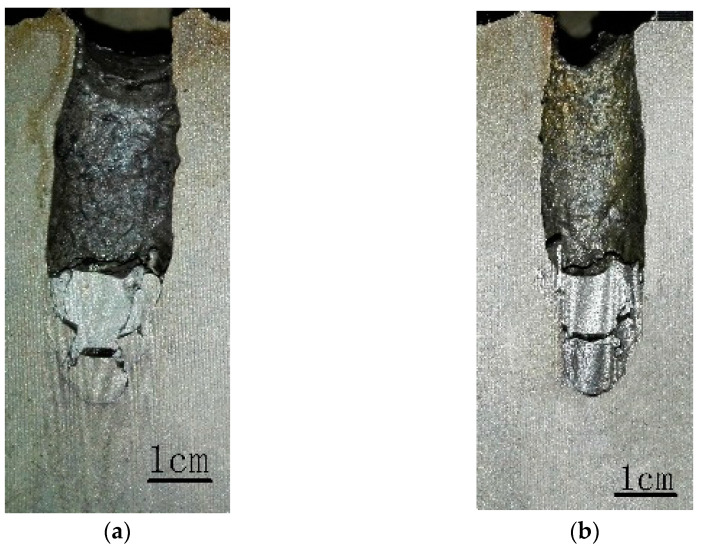
Longitudinal sections of craters penetrated by WHA rods. (**a**) *v* = 1447 m/s, (**b**) *v* = 1604 m/s.

**Figure 6 materials-16-00040-f006:**
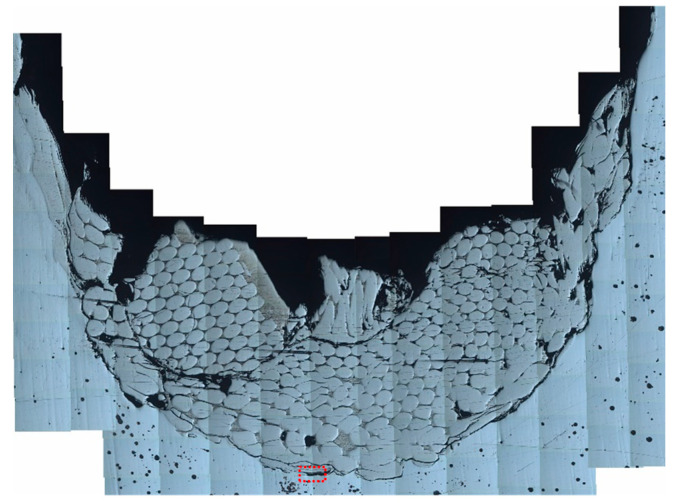
Metallographic photo of residual WF/Zr-MG rod (No. C2) at 1523 m/s.

**Figure 7 materials-16-00040-f007:**
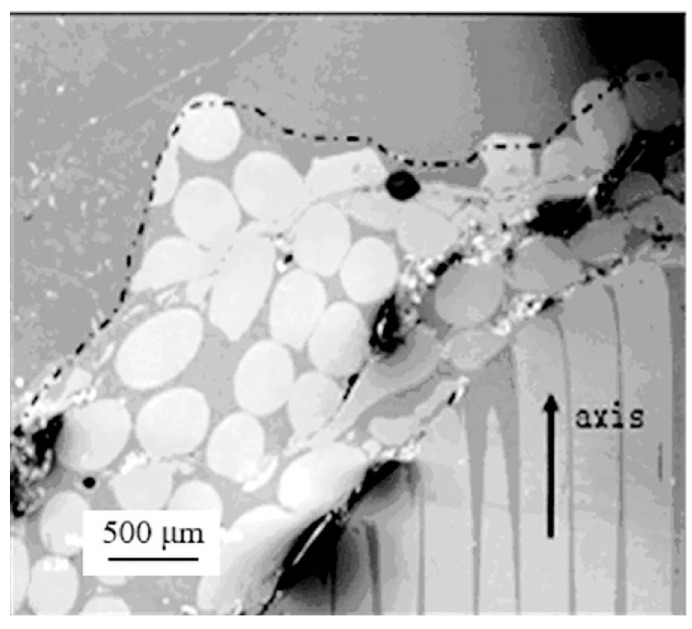
Metallographic photo of residual WF/Zr-MG rod at 811.2 m/s [16].

**Figure 8 materials-16-00040-f008:**
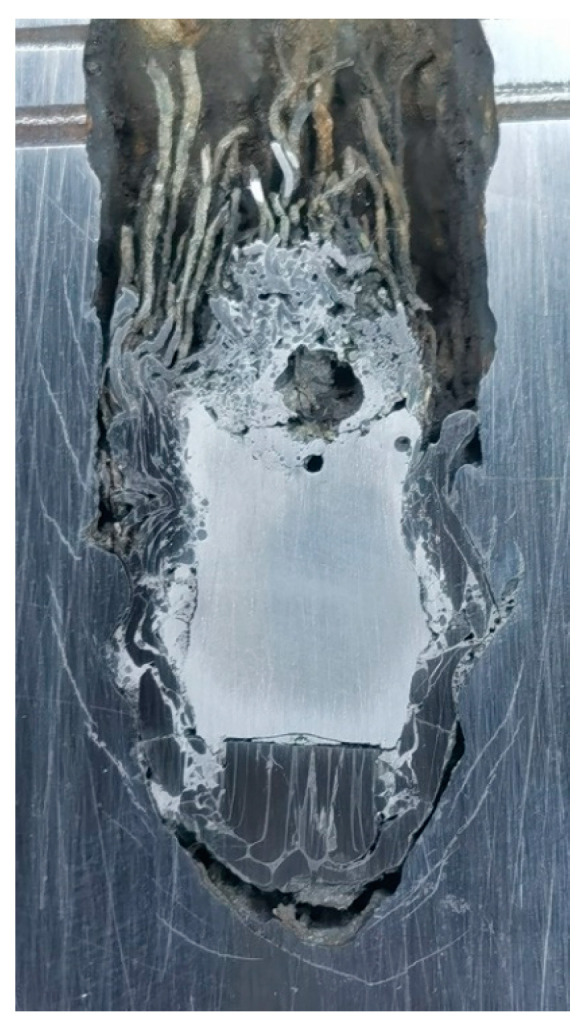
Bending and backflow fracture mode of tungsten fibers during penetration of WF/Zr-MG rod [20].

**Figure 9 materials-16-00040-f009:**
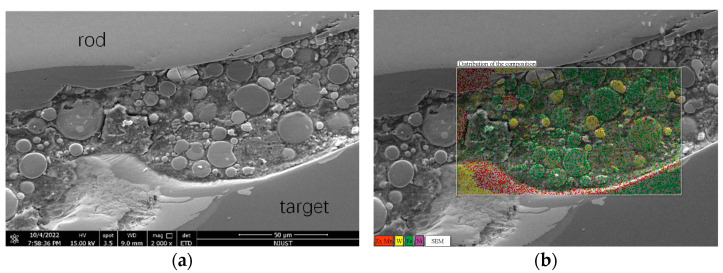
SEM image of the interface of the rod and target. (**a**) SEM image of the interface; (**b**) Composition of the material.

**Figure 10 materials-16-00040-f010:**
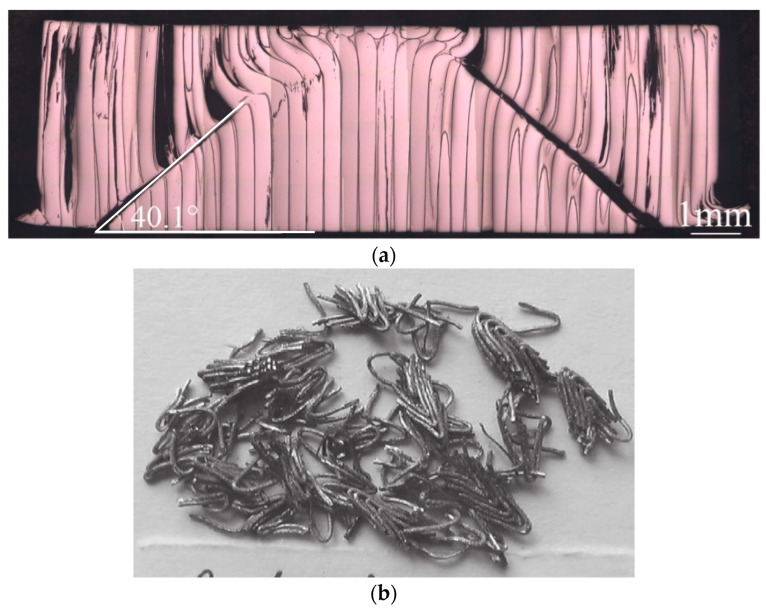
Deformation modes of WF/Zr-MG under dynamic compression [4,5]. (**a**) Under room temperature; (**b**) Under 773 K.

**Figure 11 materials-16-00040-f011:**
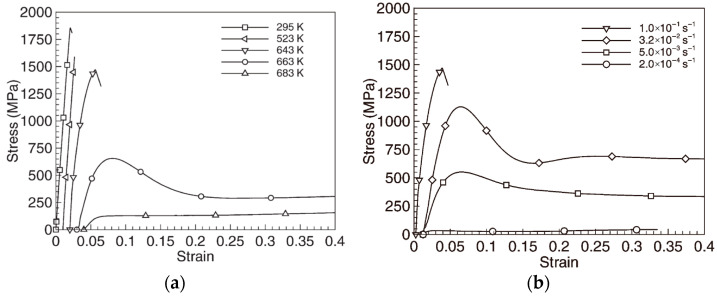
Compressive stress–strain curves of Zr-MG at (**a**) different temperatures and (**b**) different strain rates [21].

**Figure 12 materials-16-00040-f012:**
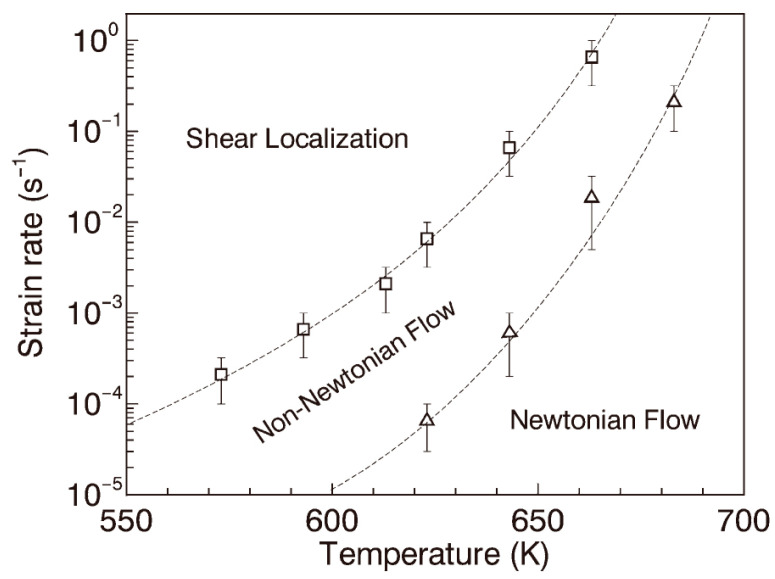
The boundaries between the three distinct deformation modes of Zr-MG [21].

**Figure 13 materials-16-00040-f013:**
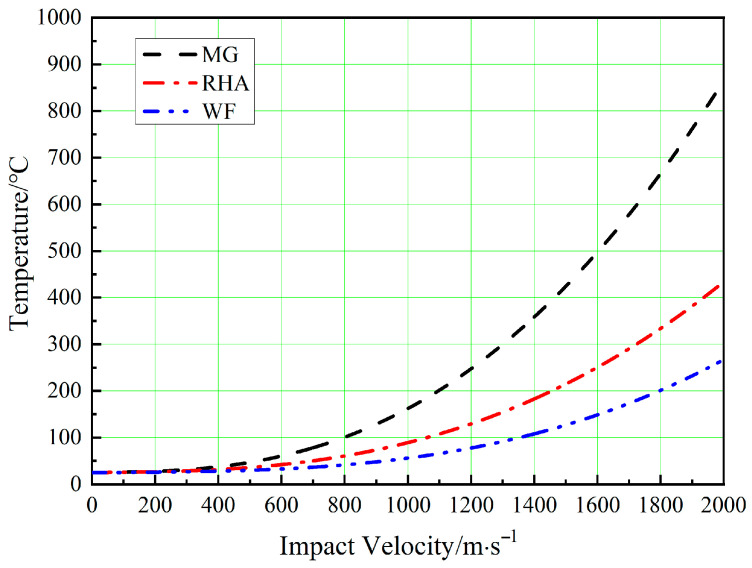
Relationship between temperature and impact velocity of Zr-MG, tungsten fibers, and RHA when WF/Zr-MG rods penetrate RHA.

**Table 1 materials-16-00040-t001:** Masses of projectiles and rods.

No.	Material	Mass of Projectile (g)	Mass of Rod (g)
C1	WF/Zr-MG	112	62
C2		113	62
W1	WHA	114	63
W2		115	63

**Table 2 materials-16-00040-t002:** Dimensions of the craters.

No.	Material	Impact Velocity (m/s)	DOP (mm)	Length of TEP (mm)	Length of QSPP (mm)	Length of SPP (mm)	Diameter of QSPP (mm)
C1	WF/Zr-MG	1560	60	12.0	37.3	10.6	16.8
C2		1523	63	10.1	43.3	9.6	16.3
W1	WHA	1447	56	13.6	22.6	19.8	19.6
W2		1604	58	14.5	23.6	20.0	18.7

**Table 3 materials-16-00040-t003:** Shock and thermodynamic properties of RHA, WF/Zr-MG, Zr-MG, and tungsten fiber [25,26].

Material	$\rho_{0}$ (kg/m^3^)	$V_{0}$ (m^3^/kg)	$C_{0}$ (m/s)	S	$C_{v}$ (J/kg·K)	$\gamma_{0}$
RHA	7850	1.27 × 10^−4^	4570	1.49	477	1.65
WF/Zr-MG	17,000	5.88 × 10^−5^	4559	1.25		
Zr-MG	6600	1.52 × 10^−4^	4136	1.29	400	1.08
Tungsten fiber	19,220	5.20 × 10^−5^	4040	1.24	134	1.58

## Data Availability

Not applicable.

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
