# Peer review of "Penetration Fracture Mechanism of Tungsten-Fiber-Reinforced Zr-Based Bulk Metallic Glasses Matrix Composite under High-Velocity Impact"

_materials, 2022, doi:10.3390/ma16010040_

Round 1

Reviewer 1 Report

This manuscript reveals the penetration fracture mechanism of tungsten fiber-reinforced Zr-based bulk metallic glasses matrix composite rods when subjected to high-velocity impact. A theoretical calculation was also made to reveal the penetration fracture mechanism. Although this manuscript has some value, it does not attain the publishable standard. The scope of this research study is too simplistic. Therefore, it is hard to appreciate the contribution of this work to the research community. It should be better developed to meet the desired level of scientific merit.

1. The majority of the references are too old. Please refer to and add some recent research articles which are related to this work.

2. In the introduction, please add more recent literature studies.

3. On page 2; section 2, how did the authors prepare the WF/Zr-Mg rod? What are the sources/suppliers of all the materials used in this research study?

4. On page 5; the second paragraph of section 3, please rewrite these sentences ‘Figure 6 is metallographic photo of……obtained Rong et al. [10].’

5. On page 7, the caption of Figure 9 should be modified. The authors have mentioned Figure 9(a) and Figure 9(b) in the text, but they are not mentioned in the caption of Figure 9.

6. On page 7; the first paragraph, ‘The experimental result has proved…..’ should be combined with the previous paragraph since this paragraph has only one sentence.

Reviewer 2 Report

The authors present an experimental study of penetration fracture mechanism of metallic glasses matrix composite. The work is interesting and performed using a variety of experimental techniques, which assumes the conclusions made are solid. Thus, the manuscript is worth to be published after a minor revision. My specific comments are the following:

1.     The authors performed a good and exhaustive instrumental study of the penetration fracture mechanism, but the work will only benefit if the authors add a short comment on the possibility of theoretical modelling of stress propagation and fracture mechanism. This can inspire other theoreticians to contribute to the field.

2.     Based on the previous comment, I suggest the authors citing the following paper (https://doi.org/10.1016/j.fpc.2021.02.007), which demonstrates how similar studies can be performed in silico.

3.     The authors often put the dimension and the value without spacing (e.g., 1000m/s). Perhaps these are typos. But it shouldn't be.

Author Response

The paper has been revised according to the comments of the reviewer. The details are as follows:

Comment 1: The authors performed a good and exhaustive instrumental study of the penetration fracture mechanism, but the work will only benefit if the authors add a short comment on the possibility of theoretical modelling of stress propagation and fracture mechanism. This can inspire other theoreticians to contribute to the field.

Answer: Previous dynamic compression tests of WF/Zr-MG have shown that the stress propagation is also a factor affecting its deformation mode. However, in the process of penetration, impact velocities and impact energy are much higher than those of dynamic compression test, which directly leads to the shear failure of WF/Zr-MG. There-fore, the propagation of stress wave is no longer an important influence factor of penetration fracture mode.

Comment 2: Based on the previous comment, I suggest the authors citing the following paper (https://doi.org/10.1016/j.fpc.2021.02.007), which demonstrates how similar studies can be performed in silico.

Answer: I am very sorry for not citing the paper (https://doi.org/10.1016/j.fpc.2021.02.007) in the article. I have read the paper, but there is no description about the fracture mechanism of silicon in the article, so I did not cite this article.

Comment 3: The authors often put the dimension and the value without spacing (e.g., 1000m/s). Perhaps these are typos. But it shouldn't be.

Answer: The mistakes are have been corrected.

Reviewer 3 Report

After a complete evaluation of the manuscript “Penetration Fracture Mechanism of Tungsten Fiber Reinforced Zr-Based Bulk Metallic Glasses Matrix Composite under High-Velocity Impact”,  The manuscript is interesting. However, it needs some changes before further processing

1.      The research gap and novelty must be highlighted in the abstract and introduction of the current work.

2.      While reading throughout the manuscript there are some typos and grammatical errors. Complete proofreading is required

3.      It is suggested to focus the conclusion according to the objective of the current work.

4.      The experiments and discussions performed must be directed toward the targeted application.

Author Response

The paper has been revised according to the comments of the reviewer. The details are as follows:

Comment 1: The research gap and novelty must be highlighted in the abstract and introduction of the current work.

Answer: The abstract and introduction are revised to add the research gap and novelty.

Comment 2: While reading throughout the manuscript there are some typos and grammatical errors. Complete proofreading is required

Answer: The typos and grammatical errors have been corrected.

Comment 3: It is suggested to focus the conclusion according to the objective of the current work.

Answer: The conclusion is rewritten.

Comment 4: The experiments and discussions performed must be directed toward the targeted application.

Answer: According to the research targeted, the title in the discussion is modified.

Round 2

Reviewer 1 Report

The authors have amended the manuscript based on the comments given by the reviewer. This manuscript looks much better than the previous version. Therefore, I recommend the acceptance of the manuscript for publication.